# PI3K and PINK1 Immunoexpression as Predictors of Survival in Patients Undergoing Resection of Brain Metastases from Lung Adenocarcinoma

**DOI:** 10.3390/ijms26072945

**Published:** 2025-03-24

**Authors:** Miriam Rubiera-Valdés, Mª Daniela Corte-Torres, Andrea Navarro-López, Noelia Blanco-Agudín, Santiago Fernández-Menéndez, Kelvin M. Piña-Batista, Jorge Santos-Juanes, Jesús Merayo-Lloves, Luis M. Quirós, Adela A. Fernández-Velasco, Iván Fernández-Vega

**Affiliations:** 1Department of Pathology, Central University Hospital of Asturias (HUCA), 33011 Oviedo, Spain; miriamrubiera@gmail.com; 2Biobank of Principality of Asturias (BioPA), 33011 Oviedo, Spain; mdanielac@hotmail.com (M.D.C.-T.); andreanavarrolopez@gmail.com (A.N.-L.); sfmenendez@gmail.com (S.F.-M.); 3Health Research Institute of the Principality of Asturias (ISPA), 33011 Oviedo, Spain; blancoanoelia@uniovi.es (N.B.-A.); merayojesus@uniovi.es (J.M.-L.); quirosluis@uniovi.es (L.M.Q.); 4Department of Functional Biology, University of Oviedo, 33006 Oviedo, Spain; 5Instituto Universitario Fernández-Vega, Universidad de Oviedo, 33012 Oviedo, Spain; 6Department of Neurology, Central University Hospital of Asturias (HUCA), 33011 Oviedo, Spain; 7Department of Neurosurgery, Central University Hospital of Asturias (HUCA), 33011 Oviedo, Spain; pineappledr@gmail.com; 8Department of Dermatology, Central University Hospital of Asturias (HUCA), 33011 Oviedo, Spain; jorgesantosjuanes@gmail.com; 9Department of Surgery and Medical-Surgical Specialties, University of Oviedo, 33006 Oviedo, Spain

**Keywords:** PI3K, PINK1, adenocarcinoma, brain, lung, immunohistochemistry, metastasis, surgery

## Abstract

Phosphoinositide 3-kinase (PI3K) and PTEN-induced kinase 1 (PINK1) are key regulators of metabolism and mitochondrial quality control. This study assessed their immunoexpression in 22 patients with lung adenocarcinoma and resected brain metastases who underwent curative treatment between 2007 and 2017 and evaluated their prognostic significance. Tissue microarrays of primary tumors and matched metastases were analyzed using the H-score method. PI3K expression was significantly higher in primary tumors (96.8 ± 57.9 vs. 43.5 ± 62.3; *p* = 0.003) and in stage IV adenocarcinomas (113.3 ± 56.3 vs. 61.4 ± 47.1; *p* = 0.043). PINK1 expression showed no significant variation across disease stages. Univariate analysis identified older age (>55 years), PI3K overexpression (HR = 7.791, 95% CI 1.718–36.432; >50 points), and PINK1 overexpression (>100 points) in primary tumors as predictors of poor overall survival (HR = 2.236, 95% CI 1.109–4.508; *p* = 0.025). Multivariate analysis confirmed PINK1 overexpression in primary tumors as an independent prognostic factor (HR = 4.328, 95% CI 1.264–14.814; *p* = 0.020). These findings suggest that PI3K and PINK1 may serve as prognostic biomarkers in lung adenocarcinoma with resected brain metastases, emphasizing the need for research on their role in tumor progression and therapeutic response.

## 1. Introduction

Lung cancer constitutes a major global health challenge due to its high incidence and mortality rates, remaining the leading cause of cancer-related deaths worldwide [1]. Despite advancements in diagnostic techniques and treatments, lung cancer is often identified at advanced stages, significantly restricting therapeutic options and worsening prognosis [2]. Among its histological subtypes, lung adenocarcinoma stands out as the most prevalent, representing approximately 40% to 50% of cases, and is increasingly diagnosed in individuals without a history of smoking, particularly women in regions such as East Asia, frequently driven by oncogenic mutations involving EGFR [3,4,5]. Brain metastases (BM) occur in about 30% of patients with non-small cell lung cancer (NSCLC), profoundly affecting patient outcomes due to limited therapeutic effectiveness and reduced survival. [6]. Nearly half of these cases present brain involvement at initial diagnosis, with the rest developing metastases during disease progression [7]. Effective management of oligometastatic brain lesions through surgery, radiation, targeted systemic therapies and immune checkpoint inhibitors is crucial for symptom control and survival improvement [8,9]. Nonetheless, prognosis remains poor, underscoring the urgent need for novel prognostic biomarkers and therapeutic targets [10,11].

Phosphoinositide 3-kinase (PI3K) is a critical component of the PI3K/AKT/mTOR signaling pathway, frequently activated in various malignancies [12]. Aberrant activation of PI3K contributes to oncogenic processes, including cell proliferation, survival, angiogenesis, and metabolic adaptation [13]. In lung adenocarcinoma, PI3K pathway alterations have been associated with poor prognosis and therapeutic resistance, making this pathway a promising target under active investigation, with several inhibitors currently in clinical trials [14,15,16].

Mitophagy, a selective form of autophagy, is responsible for eliminating damaged mitochondria and plays an important homeostatic function in cells and tissues, maintaining the integrity of the mitochondrial pool by eliminating old and/or damaged mitochondria [17,18]. Consequently, defects in mitophagy could lead to a failure in the proper reprograming of cellular metabolism, the control of cell fate determination, the attenuation of inflammation, and response to DNA damage [18]. PTEN-induced kinase-1 (PINK1) is a key regulator of the canonical mitophagy pathway, initiating the process by recruiting Parkin, an E3 ubiquitin ligase, to the outer mitochondrial membrane [19]. PINK1 signaling has been shown to modulate several cellular processes, including mitochondrial dynamics, bioenergetics, and oxidative stress responses [20]. Dysregulation of mitophagy has been implicated in cancer, with PINK1 playing a dual role, depending on the cellular context. Thus, PINK1 may act either as a tumor suppressor by maintaining mitochondrial integrity or as a promoter of tumorigenesis through metabolic reprogramming and apoptosis resistance [21,22,23,24].

Recent evidence highlights critical interactions between the PI3K/AKT/mTOR pathway and PINK1, suggesting their crosstalk significantly impacts cancer progression, therapeutic resistance, and metabolic plasticity, especially under stress conditions such as metastatic dissemination to the brain. Moreover, their interactions may modulate the tumor microenvironment, influencing immune responses and inflammation, which are established hallmarks of cancer [25,26,27]. For instance, PINK1 enhances AKT activity by regulating PTEN, the main inhibitor of the PI3K/AKT/mTOR pathway [24,26]. Additionally, PINK1 has been associated with glycolysis regulation and modulation of PI3K signaling, promoting cancer cell survival and proliferation [28,29]. Moreover, PINK1 and PI3K may influence the tumor microenvironment, particularly through their interactions with immune cells and their role in controlling the inflammatory response [30,31]. These findings underscore the potential of PINK1 and PI3K as prognostic biomarkers and therapeutic targets.

In this study, we aimed to investigate PI3K and PINK1 immunoexpression in lung adenocarcinomas with resected brain metastases and assess their potential prognostic significance. We also explored the association of PI3K and PINK1 expression with clinicopathological features, patient outcomes, and PD-L1 immunoexpression.

## 2. Results

### 2.1. Clinicopathological Features of Cases

Twenty-two consecutive patients with lung adenocarcinoma and BM, for whom suitable tumor material was available, were enrolled in the study. Patients were categorized into two groups based on the presence of brain metastases at diagnosis: stage IV or earlier stages, according to their main clinicopathological features, as summarized in Table 1. Except for survival, adjuvant therapy, PI3K expression in the primary tumor, and PDL1 expression in BM, there were no significant differences between the two groups regarding clinicopathological characteristics. Patients in Stages I–III showed significantly longer overall survival compared to those in Stage IV (*p* = 0.031). Adjuvant therapy was more frequently administered in Stage IV patients (*p* = 0.014). The expression of PI3K in primary tumors was significantly higher in Stage IV cases (*p* = 0.043) and was more than twice as high as in BM samples (*p* = 0.003). Additionally, PDL1 expression in BM was significantly different between the groups (*p* = 0.035). No significant differences were observed regarding age, gender, primary tumor location, BM location, tumor size, differentiation grade, necrosis, and mitotic activity.

Several significant correlations were identified between clinicopathological variables in lung adenocarcinoma and its BM (Table 2). A significant moderate negative correlation between OS and patient age was noted (r = −0.590; *p* = 0.016), indicating that older patients had shorter OS. A strong positive correlation was observed between mitotic activity in the primary tumor and tumor size (r = 0.673; *p* = 0.004), suggesting that larger tumors exhibited higher mitotic activity. Furthermore, mitotic activity in brain metastases showed a significant moderate positive correlation with mitotic activity in the primary tumor (r = 0.594; *p* = 0.015), highlighting a link between proliferative activity in both sites. Additionally, PI3K expression in the primary tumor displayed a significant positive correlation with PI3K expression in BM (r = 0.556; *p* = 0.025), suggesting a consistent activation pattern of this pathway in primary and metastatic lesions.

### 2.2. Immunohistochemical Study of Proteins

A mean immunoexpression of 96.8 ± 57.9 points for PI3K in primary lung adenocarcinomas and 43.5 ± 62.3 points in BM was determined, with statistically significant differences between both tumor sites (*p* = 0.003) (Table 1). Cytoplasmic staining with variable nuclear positivity for PI3K was observed among tumor samples (Figure 1A–C). Regarding PINK1, no significant differences were found between primary tumors (76.8 ± 40.0) and BM (77.5 ± 44.8; *p* = 0.793) (Table 1). The immunostaining pattern for PINK1 varied among tumor samples, showing weak to moderate or intense cytoplasmic positivity (Figure 1D–F). These findings indicate a significant reduction of PI3K expression in metastatic lesions, while PINK1 levels remain stable regardless of the tumor site.

PD-L1 expression was evaluated in both primary lung adenocarcinomas and their corresponding BM. Positive immunostaining (PD-L1 > 1%) was detected in 40.9% of primary tumors and 36.4% of BM, without statistically significant differences between both sites (*p* = 0.307). In addition, no positive cases were identified in BM from Stages I–III lung adenocarcinomas, with significant differences compared to Stage IV cases (*p* = 0.035) (Table 1). Appendix A shows images of the immunohistochemical analysis.

### 2.3. Survival Curves, and Univariate and Multivariate Analysis

Kaplan–Meier survival analysis was performed to evaluate the prognostic impact of PI3K PINK1 and PD-L1 expression on OS in lung adenocarcinoma patients with BM. The analysis was conducted in both primary tumors and metastatic lesions; however, significant results were only observed in the primary tumors. The results demonstrated that patients with more than 50 points of PI3K expression in the primary tumor exhibited significantly reduced OS compared to those with lower PI3K levels (*p* = 0.002, Figure 2A). Regarding PINK1 expression, patients with more than 90 points in the primary tumor had significantly lower OS compared to those with 90 or fewer points (*p* = 0.044, Figure 2B). When stratified into three groups, patients with PI3K expression between 60 and 150 points had the worst OS, followed by those with more than 150 points (*p* = 0.008, Figure 2C). Similarly, when PINK1 expression was categorized into three groups, patients with more than 100 points were significantly associated with poorer survival outcomes (*p* = 0.049, Figure 2D). These findings suggest that both PI3K and PINK1 overexpression in primary tumors may be linked to worse prognosis in lung adenocarcinoma patients with BM. Kaplan–Meier survival analysis was also performed to assess the prognostic significance of PD-L1 expression. The results indicated that patients with PD-L1–positive tumors had significantly reduced OS compared to those with PD-L1–negative tumors (*p* = 0.048, Appendix A). This finding suggests that PD-L1 positivity may be associated with worse prognosis in this patient cohort.

Univariate analysis showed that older age (>55 years) (HR = 7.014, 95% CI 1.485–33.139; *p* = 0.014), more than 50 points of PI3K expression in the primary tumor (HR = 7.791, 95% CI 1.718–36.432; *p* = 0.008), and more than 100 points of PINK1 expression in the primary tumor (HR = 2.236, 95% CI 1.109–4.508; *p* = 0.025) were significantly associated with poor OS. Additionally, in the multivariate model, PINK1 overexpression in the primary tumor remained an independent prognostic factor for OS (HR = 4.328, 95% CI 1.264–14.814; *p* = 0.020) (Table 3).

## 3. Discussion

Lung cancer is the leading cause of cancer-related mortality worldwide and remains a major public health concern [1]. Lung adenocarcinoma, like other types of lung cancer, is often diagnosed at a locally advanced or advanced stage, which limits treatment options [2].

Therefore, disease stage is a key prognostic factor, significantly impacting survival outcomes [32]. In particular, lung adenocarcinoma with brain metastases has historically posed a significant challenge in oncology due to its poor prognosis, complex molecular landscape, and the difficulty of its treatment [6]. In fact, in this case series, for patients diagnosed at stage IV, surgical management always prioritized the resection of brain metastases before addressing the primary tumor [8]. In this work, we studied the immunohistochemical expression of the kinases PI3K, PINK1, and PD-L1 in a specific subset of patients with lung adenocarcinomas—only those who underwent surgical resection of their BM—to better describe molecular alterations at this stage of advanced disease.

We stratified our twenty-two cases by disease stage and identified relevant findings. Patients diagnosed at earlier stages (I–III) had more than twice the OS compared to those diagnosed at Stage IV. A recent multicenter study in China analyzing 7311 lung cancer patients confirmed that disease stage is a crucial predictor of survival, with significant differences in 5-year overall survival rates across Stages I to IV. This highlights the impact of early-stage diagnosis and suggests that aggressive screening strategies may benefit high-risk populations, such as heavy smokers, to improve patient outcomes [33]. Adjuvant therapy was administered to most patients in this cohort, with a statistically significant difference between Stage IV and Stage I–III cases. We observed that patients with advanced-stage disease are more likely to receive adjuvant therapy, likely reflecting the greater tumor burden and the necessity of multimodal approaches in Stage IV disease. Previous studies have demonstrated that combining systemic therapy with RT can improve survival in NSCLC patients with brain metastases [34]. The higher percentage of patients in Stages I–III not receiving adjuvant therapy (42.8%) is due to surgical resection being the primary treatment. Furthermore, the rise of targeted therapies and immune checkpoint inhibitors is reshaping treatment paradigms, particularly in patients with specific molecular alterations [11].

Microscopic examination of the tumors did not show significant differences concerning grade and necrosis. However, Stage IV tumors exhibited higher mitotic activity in both primary and metastatic sites compared to Stage I–III tumors. The direct relationship between mitotic activity and Stage IV lung adenocarcinoma is not clearly established in the current literature. Although it is reasonable to assume that tumors in advanced stages may exhibit higher mitotic activity due to their aggressive nature, additional studies are needed to confirm this specific association. Unlike mitotic count, the tumor proliferative marker Ki-67 has been associated with a poorer prognosis in lung cancer [35]. The PI3K H-score showed a statistically significant increase, nearly doubling from Stage I–III tumors to Stage IV tumors. It is known that the activation of the PI3K/AKT pathway acts as an inducer of the epithelial-mesenchymal transition process during lung tumor metastasis [36]. In addition, the PI3K H-score showed a statistically significant decrease of more than one half from primary tumors to brain metastases. This reduction may indicate a biological shift in metastatic lesions that could influence therapeutic resistance, potentially affecting the response to PI3K inhibitors [37]. PI3K has been extensively studied in various primary tumors and metastatic tissues, emerging as a master regulator of brain metastasis-promoting macrophages and microglia. It plays a pivotal role in shaping the tumor microenvironment, thereby facilitating metastatic colonization within the central nervous system [24,38]. Additionally, patients with Stage IV tumors exhibited higher PD-L1 expression compared to Stage I–III cases, especially in brain metastases, which may indicate a more immunosuppressive tumor microenvironment, facilitating tumor progression and immune evasion. This suggests that tumors diagnosed at an advanced stage exhibit a more pronounced immune evasion phenotype, potentially affecting their response to immune checkpoint inhibitors (ICIs) [39]. Concerning PINK1 immunohistochemical expression, we did not observe significant differences between groups, including disease stages and primary tumors, compared to BM. However, we previously reported significantly higher levels of PINK1 immunoexpression in liver metastases from colorectal carcinomas compared to primary tumors of the colon [22]. In addition, a current pan-cancer analysis revealed that PINK1 mRNA expression was reduced across multiple cancer types compared to normal tissues, including brain, breast, colorectal, esophageal, head and neck, liver, and ovarian cancers, as well as leukemia and melanoma, while elevated expression was observed in diffuse large B-cell lymphoma [30]. In NSCLC, the observed PINK1 expression pattern in our study is generally consistent with previous findings in lung cancer research. Thus, PINK1 exhibited significant overexpression in both NSCLC tissues and cell lines, showing a correlation with the clinicopathological features of the disease [40]. Building upon these findings, we further investigated PINK1 expression by directly comparing its levels in tumor tissues from both primary and metastatic lesions, aiming to explore its potential applicability in routine pathological practice.

Although PINK1 inhibits PTEN, the key inhibitor of the PI3K/AKT pathway, we did not observe any positive correlations in the immunohistochemical expression between the two molecules in either primary tumors or BM. This suggests the presence of more complex regulatory mechanisms that may attenuate this effect [41].

Survival analysis revealed that PI3K and PINK1 expression levels in primary lung adenocarcinomas were significantly associated with OS when patients were stratified according to specific cutoff points. These findings were also consistent with univariate Cox regression analyses. In contrast, we previously observed that immunohistochemical expression in metastatic tissue was not significantly associated with survival [22]. Furthermore, PINK1 expression in primary tumors emerged as an independent predictor of OS in the multivariate model. Similar findings were reported by Meng Wang et al. in primary lung adenocarcinomas [42]. Moreover, we previously reported analogous findings in metastatic colorectal carcinomas with resected liver metastases [22]. Additional clinical variables, particularly ECOG performance status and systemic treatment response, could potentially influence our prognostic analysis. In addition, another relevant clinical variable, such as extracranial disease burden, was intentionally excluded, since all patients in our cohort were negative for extracranial tumor involvement. In this context, the molecular profile of advanced-stage cancer is becoming increasingly relevant, in some cases surpassing conventional clinical and histopathological factors. Taken together, molecular findings related to PI3K and PINK1 could be integrated into prognostic models, providing a comprehensive and surgery-specific approach for stratifying patients with brain metastases [43,44].

While the role of PI3K and PINK1 in various cancer types is still not fully understood, previous studies have shown that PI3K acts as a master regulator of brain metastasis, facilitating metastatic colonization. On the other hand, PINK1 has shown context-dependent effects, promoting cell migration and proliferation in lung cancer while exhibiting a protective role in other malignancies such as blood, brain, and breast cancers [30,42]. Altogether, these findings suggest that PI3K and PINK1 alterations are associated with carcinogenesis, particularly in lung adenocarcinomas with BM, potentially due to their roles in mitophagy, metabolic reprogramming, and modulation of the tumor microenvironment. From a translational perspective, therapies targeting these pathways, such as PI3K inhibitors or mitophagy modulators like metformin, may hold promise for improving patient outcomes, particularly in those with lung adenocarcinoma and BM exhibiting high PI3K and PINK1 expression [45,46].

We acknowledge several limitations in our study. First, the retrospective nature of the study inherently introduces potential biases. These include selection bias, as the inclusion of patients was based on available medical records and tumor tissue, which may not fully represent the entire patient population. Information bias is also possible due to inaccuracies or incompleteness in the recorded data. Additionally, retrospective studies are limited in controlling for confounding variables, which may influence the observed associations. Second, the study is limited to a highly specific subset of lung cancer patients, resulting in a small sample size, and we did not evaluate protein immunoexpression in non-metastatic lung adenocarcinomas with comparable characteristics. Third, these findings are based on patients treated at a university hospital, thus, the proportion of tumors with poor prognosis might be higher due to referral bias compared to routine ambulatory practice. Fourth, the study was conducted at a single center. Fifth, the immunohistochemical analysis was performed using tissue microarrays; however, protein immunoexpression patterns were relatively homogeneous and consistent across the three representative tissue cores selected from each tumor.

## 4. Materials and Method

### 4.1. Patients and Samples

A total of 22 consecutive patients with suitable lung adenocarcinoma tissue samples and resected BM, who underwent curative treatment between 2007 and 2017, were retrospectively selected from the Department of Pathology electronic database at the Hospital Universitario Central de Asturias. The follow-up period lasted until 2025. All patients were managed with conventional treatment strategies before the immunotherapy era, receiving standard approaches such as surgery, chemotherapy, and radiotherapy when indicated [47,48,49]. All the electronic medical records were reviewed to determine whether outcomes of interest occurred. All the tumors were excised with conventional surgery. In fact, for cases diagnosed at Stage IV, surgical management always prioritized the resection of brain metastases before addressing the primary tumor. Patients with partial or subtotal resections, or those with limited material, were excluded. The original archived H&E slides were reviewed, and diagnoses were established following the latest WHO guideline [50]. Information about the tumor stage was obtained from the date of the diagnosis. Clinical patient-related data were collected. Patient age was defined as the age at the time of diagnosis, either by lung or brain excisional biopsy. All patients had a history of heavy smoking, with a cumulative exposure of more than 30 pack-years. Comprehensive molecular profiling was performed, including immunohistochemistry (IHC) for relevant biomarkers and PCR-based analysis. All patients were found to be negative for alterations that would qualify them for targeted therapy. Ethics approval was obtained from the Ethics Committee of Hospital Universitario Central de Asturias (Reference No. 88/18), and the study was conducted in compliance with the Declaration of Helsinki.

### 4.2. Histopathologic Evaluation

Each sample was analyzed by two independent observers (and a third one in the case of strong disagreement), who confirmed the diagnosis and registered the following histopathologic features using hematoxylin-eosin-stained slides: degree of differentiation classified as well differentiated (1), moderately differentiated (2), and poorly differentiated (3); absence or presence and percentage of necrosis and mitotic activity by 10 high-power fields.

### 4.3. Tissue Microarray Construction

Tissue microarrays (TMAs) were constructed from tissue blocks used for routine pathological evaluation. Morphologically representative areas were selected from each individual tumor paraffin block. Areas in each case with the most representative histology to overcome tumor heterogeneity were selected, and three 3 mm tissue cores were taken from each donor block and extruded into the recipient array. Thus, TMAs were created containing three tissue cores from each of the 22 lung adenocarcinomas and their respective 22 BM. In addition, each TMA included two cores of normal placenta and cerebral tissue as internal controls. A section from each microarray was stained with H&E to check the adequacy of tissue sampling. After 5 min at 60 °C, the TMA blocks were subsequently cut using a microtome into 3 μm thick sections and mounted on glass slides in preparation for immunohistochemistry.

### 4.4. Immunohistochemistry

For expression analysis by immunohistochemistry, we used the EnVision FLEX High pH (Link) Kit (Agilent-Dako, K800021, Santa Clara, CA, USA) and Dako Autostainer system. Paraffin-embedded tissue sections (3 µm) were deparaffinized and rehydrated, and epitope retrieval was conducted by heat induction (HIER) at 95 °C for 20 min and at pH 9 (Agilent-Dako) in the Pre-Treatment Module, PT-LINK (Agilent-Dako). Endogenous peroxidase activity was blocked with EnVision™ FLEX Peroxidase-Blocking Reagent (DM821) for 5 min. The sections were incubated with rabbit Anti-PI3K monoclonal antibody (Cell Signaling Technology, ref: 4249S, Danvers, MA, USA) at 1:50 dilution for 30 min; rabbit Anti-PINK-1 policlonal antibody (BC100-494, NobusBiologicals, Madrid, Spain) at 1:200 dilution for 30 min; and mouse Anti–PD-L1 monoclonal antibody (22C3, Dako, Denmark), at 1:200 dilution for 30 min. The antigen–antibody reaction was detected with the Dako EnVision + Dual Link System-HRP (Agilent-Dako). The signal was detected using diaminobenzidine chromogen as a substrate in Dako EnVision™ FLEX/HRP (Agilent-Dako). Counterstaining with hematoxylin was the final step. Negative controls were processed by omitting the primary antibody. Normal placental and cerebral tissues were used as positive controls. After the whole process, sections were dehydrated and mounted with permanent medium (Agilent-Dako mounting medium, CS703). The sections were studied and photographed under a light microscope (Nikon—Eclipse 80i, Nikon Corporation, Tokyo, Japan).

### 4.5. Immunohistochemistry Assessment

Immunoexpression of the proteins was evaluated by two independent observers (and a third one in the case of strong disagreement) without any prior knowledge of each patient’s clinical information and outcome. We used a semiquantitative approach called H-score (or “histo” score), as described elsewhere [22,51,52]. The final score gives more relative weight to higher intensity staining in each tumor sample. Then, the sample can be categorized using a qualitative variable and considered positive or negative based on a specific discriminatory threshold. In addition, the final score was the mean of three cores analyzed for each case. In addition, excellent agreement was obtained for the immunohistochemistry assessment by the observers (κ = 0.783; 95% CI: 0.672–0.894), based on a hierarchical kappa test. Strong disagreement was considered for those cases evaluated with more than 30 points of difference. Discrepant cases were reevaluated, and the disagreement resolved.

### 4.6. Statistical Analysis

Baseline demographic and clinical characteristics of the patients and pathological data were summarized with standard descriptive statistics. The primary endpoint analyzed was overall survival (OS), defined as the time from the date of diagnosis confirmed by excisional biopsy to the date of death or the last recorded follow-up. All deaths were tumor related. All parameters were tested for normal distribution by the Shapiro–Wilk test. Therefore, depending on their symmetry and nature, variables were described using mean ± standard deviation, percentage, medians with 25 and 75 percentiles, or relative and absolute frequencies. The association between categorical variables was analyzed using the χ^2^ test. For statistical analysis involving quantitative variables, non-parametric tests such as the Kruskal–Wallis test (the nonparametric version of the ANOVA) were conducted. Weighted Bonferroni correction was performed in the event of multiple statistical tests. The Dunn test was performed as a post hoc test after a significant Kruskal–Wallis test. Pearson’s correlation test was carried out to analyze the statistical relationship, or association, between two continuous variables. For analysis of the survival data of patients, the Kaplan–Meier curves and the log-rank test were performed. ROC curve analysis was performed to determine the optimal cut-off point for the H-score. The cut-off was selected by maximizing sensitivity and specificity, defined as the highest Youden index. The Cutoff Finder tool was also employed to determine the optimal cut-off values [53]. Crude and adjusted hazard ratios (HRs) and 95% confidence intervals (CIs) were calculated by using the Cox proportional-hazards model. The simultaneous prognostic effect of various factors was determined in a multivariate analysis by using the Cox proportional-hazard regression model. All reported *p* values are two-sided, and values below 0.05 are considered statistically significant. All analyses were made by using IBM SPSS Statistics version 27, IBM Corp., Armonk, NY, USA, (institutional license from the University of Oviedo).

## 5. Conclusions

In conclusion, the data presented herein are derived from a highly specific subset of patients with lung adenocarcinomas lacking major driver mutations and with resected BM; the data revealed that the immunohistochemical expression of PINK1, evaluated using the H-score and stratified into three groups (<50, 50–100, and >100), was a significant independent prognostic predictor for OS when assessed in primary tumors but not in metastatic tissue. Similarly, PI3K expression in primary tumors emerged as a significant prognostic factor, with higher H-scores associated with poorer survival outcomes. Interestingly, immunohistochemical analysis of brain metastases did not show any significant association with patient prognosis, highlighting the novelty of this finding. Thus, high PINK1 and PI3K expression in primary lung adenocarcinomas with brain metastases are associated with poorer overall survival, and their identification may assist in therapeutic decision making, guiding more aggressive treatment strategies, or closer monitoring for patients with a worse prognosis. In addition, our findings indicate a possible association between PI3K/PINK1 expression and survival in this subgroup of lung adenocarcinoma with resected brain metastases, though further studies are required to confirm their prognostic value and underlying mechanisms. These results support the potential utility of PI3K-targeted therapies and mitophagy modulators in selected patient subgroups. Moreover, PD-L1 expression reinforces the need for personalized approaches in immunotherapy. Taken together, these results require validation in prospective studies that also include lung adenocarcinomas with metastases to other anatomical sites, further supporting the clinical utility of PINK1 and PI3K immunoexpression for prognostic assessment. Finally, given their potential relevance, PI3K and PINK1 could be readily integrated into clinical practice within pathology services by incorporating them into the immunohistochemistry service portfolio, thereby facilitating their routine assessment in diagnostic workflows.

## Figures and Tables

**Figure 1 ijms-26-02945-f001:**
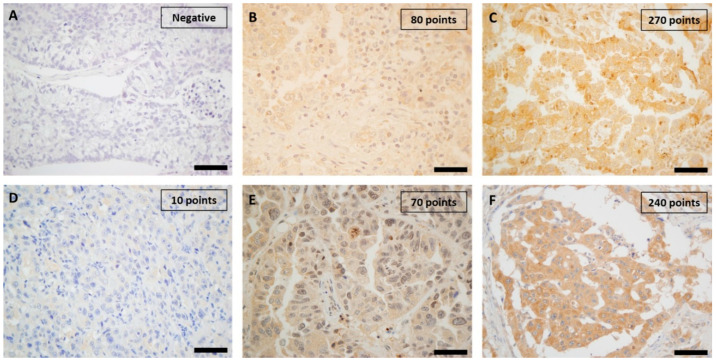
Immunostaining for PI3K and PINK1 proteins in primary lung adenocarcinomas, measured by the H-score. Images show PI3K expression (**A**–**C**) and PINK1 expression (**D**–**F**) at 400× magnification. Representative cases are shown with negative or low expression (**A**,**D**), moderate expression (**B**,**E**), and high expression (**C**,**F**). Scale bars in (**A**–**F**) = 40 μm.

**Figure 2 ijms-26-02945-f002:**
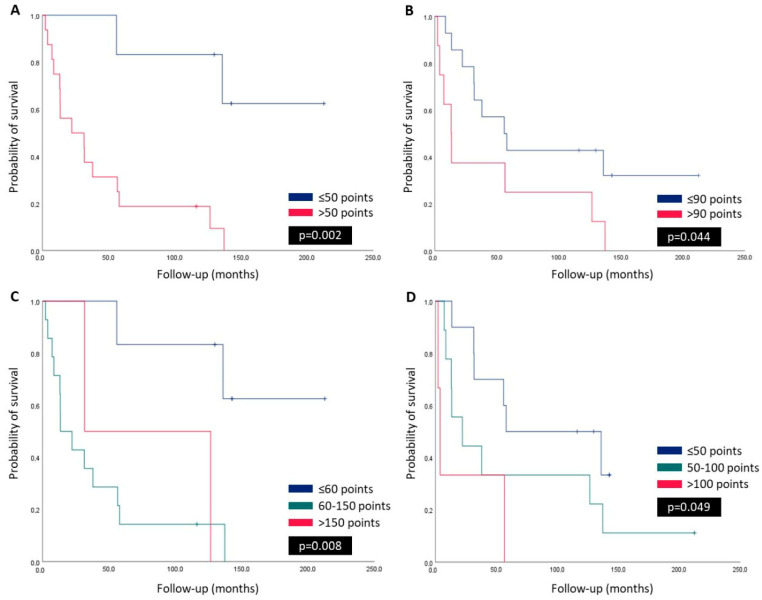
Kaplan–Meier survival estimates for PI3K and PINK1 expression in primary lung adenocarcinomas. (**A**) PI3K expression grouped by H-score values ≤50 and >50 points, showing significantly better OS for cases with lower expression (*p* = 0.002). (**B**) PINK1 expression grouped by H-score values ≤90 and >90 points, with higher survival in the low-expression group (*p* = 0.044). (**C**) PI3K expression categorized into three groups, ≤60, 60–150, and >150 points, highlighting significantly better OS for cases with H-scores ≤60 points (*p* = 0.008). (**D**) PINK1 expression stratified into three groups, ≤50, 50–100, and >100 points, showing improved survival in cases with H-scores ≤50 points (*p* = 0.049).

**Table 1 ijms-26-02945-t001:** Patient demographics and histopathological differences based on brain metastasis at diagnosis.

	Total	Stage IV	Stages I–III	*p* Value
**Patients**	22	15 (68.2%)	7 (31.8%)	-
**Age (years)** **range**	58.7 ± 8.7 36–77	60.7 ± 5.4 52–71	54.4 ± 12.7 36–77	0.113
**Gender** **male** **female**	11 (50%) 11 (50%)	8 (53.3%) 7 (46.7%)	3(42.9%) 4 (57.1%)	0.975
**Status** **alive** **deceased**	5 (22.7%) 17 (77.3%)	2 (13.3%) 13 (86.7%)	3 (42.9%) 4 (57.1%)	0.321
**Primary tumor location** **Superior lobes** **Other sites**	12 (54.5%) 10 (45.5%)	9 (60%) 6 (40%)	3 (42.9%) 4 (57.1%)	0.361
**Brain metastases location** **Fronto-temporal lobes** **Other sites**	17 (77.3%) 5 (22.7%)	11 (73.3%) 4 (26.7%)	6 (85.7%) 1 (14.3%)	0.348
**Tumor size in primary tumor (cm)**	3.5 ± 1.4	3.6 ± 1.4	3.1 ± 1.6	0.449
**Survival (months)** **OS**	68.1 ± 62.4	50.0 ± 52.5	106.9 ± 68.0	**0.031**
**Adjuvant therapy** **No** **RT and/or CT** **RT and CT**	3 (13.6%) 14 (63.7%) 5 (22.7%)	0 12 (80%) 3 (20%)	3 (42.8%) 2 (28.6%) 2 (28.6%)	**0.014**
**Grade in primary tumor** **Well** **Moderate** **Poor**	7(31.8%) 8(36.4%) 7 (31.8%)	4 (26.7%) 6 (40%) 5 (33.3%)	3 (42.8%) 2 (28.6%) 2 (28.6%)	0.741
**Grade in brain metastasis** **Well** **Moderate** **Poor**	3 (13.6%) 1 (4.6%) 18 (81.8%)	2 (13.3%) 0 13 (86.7%)	1 (14.3%) 1 (14.3%) 5 (71.4%)	0.447
**Necrosis in primary tumor** **No** **<25%** **25%/50%** **50%/75%** **>75%**	4 (18.2%) 7 (31.8%) 7 (31.8%) 3 (13.6%) 1 (4.6%)	6 (40%) 3(20%) 4 (26.6%) 1 (6.7%) 1 (6.7%)	3 (42.8%) 2 (28.6%) 1 (14.3%) 1 (14.3%) 0	0.294
**Necrosis in brain metastasis** **No** **<25%** **25%/50%** **50%/75%** **>75%**	11 (50%) 5 (22.7%) 4 (18.2%) 0 2(9.1%)	6 (40%) 3 (20%) 4 (26.6%) 0 2 (13.4%)	5 (71.4%) 2 (28.6%) 0 0 0	0.420
**Mitotic activity (per 10 HPF)** **Primary tumor** **Brain metastases**	18.0 ± 18.4 20.7 ± 16.5	22.8 ± 19.8 23.0 ± 18.6	7.7 ± 9.5 15.7 ± 10.2	0.343 **0.026** 0.285
**PI3K (H-Score):** **Primary tumor** **Brain metastases**	96.8 ± 57.9 43.5 ± 62.3	113.3 ± 56.3 54.3 ± 70.2	61.4 ± 47.1 18.3 ± 28.6	**0.003** **0.043** 0.319
**PINK1 (H-Score):** **Primary tumor** **Brain metastases**	76.8 ± 40.0 77.5 ± 44.8	79.3 ± 39.9 75.7 ± 45.4	71.4 ± 43.0 81.7 ± 47.5	0.793 0.677 0.412
**PDL1 (positive >1% tumor):** **Primary tumor** **Brain metastases**	9 (40.9%) 8 (36.4)	7(46.7%) 8(53.3)	2 (28.6%) 0	0.985 0.307 **0.035**

Values in bold are statistically significant, *p <* 0.05. OS: Overall survival; RT: Radiotherapy; CT: Chemotherapy; HPF: High power field. The underlined *p*-values represent the results of the statistical analysis comparing the variables of the primary tumor and the variables of the brain metastasis.

**Table 2 ijms-26-02945-t002:** Pearson’s correlation test for clinicopathological data and PI3K and PINK1 immunoexpresion in lung adenocarcinomas and their brain metastases.

	Age	Tumor Size	OS	Mitosis Primary Tumor	Mitosis Brain Metastases	PI3K Primary Tumor	PI3K Brain Metastases	PIKN1 Primary Tumor
**Tumor size**	r = −0.117 *p* = 0.665	-	-	-	-	-	-	-
**OS**	**r = −0.590 *p* = 0.016**	r = −0.109 *p* = 0.687	-	-	-	-	-	-
**Mitosis primary tumor**	r = −0.351 *p* = 0.183	**r = 0.673 *p* = 0.004**	r = −0.032 *p* = 0.908	-	-	-	-	-
**Mitosis brain metastases**	r = −0.486 *p* = 0.056	r = 0.173 *p* = 0.521	r = 0.193 *p* = 0.474	**r = 0.594 *p* = 0.015**	-	-	-	-
**PI3K** **primary tumor**	r = 0.402 *p* = 0.122	r = 0.041 *p* = 0.880	r = −0.468 *p* = 0.068	r = 0.179 *p* = 0.508	r = −0.009 *p* = 0.975	-	-	-
**PI3K** **brain metastases**	r = 0.013 *p* = 0.963	r = −0.180 *p* = 0.505	r = −0.289 *p* = 0.277	r = 0.107 *p* = 0.692	r = 0.246 *p* = 0.358	**r = 0.556 *p* = 0.025**	-	-
**P** **INK1 primary tumor**	r = 0.180 *p* = 0.505	r = 0.223 *p* = 0.406	r = −0.385 *p* = 0.140	r = 0.410 *p* = 0.115	r = 0.145 *p* = 0.593	r = 0.372 *p* = 0.156	r = 0.023 *p* = 0.934	-
**PINK1** **brain metastases**	r = 0.149 *p* = 0.582	r = 0.135 *p* = 0.617	r = −0.185 *p* = 0.492	r = −0.284 *p* = 0.287	r = −0.134 *p* = 0.620	r = 0.117 *p* = 0.667	r = 0.247 *p* = 0.356	r = 0.045 *p* = 0.869

Values in bold are statistically significant, *p <* 0.05; OS: Overall survival; r = Pearson’s correlation coefficient.

**Table 3 ijms-26-02945-t003:** Univariate and multivariate analyses of variables associated with overall survival.

Variables	Univariate Analysis			Multivariate Analysis		
	HR	95% CI	*p* Value	HR	95% CI	*p* Value
**Age, years**	1.074	1.014–1.138	**0.015**			
**Age, years (≤55 vs. >55)**	7.014	1.485–33.139	**0.014**	2.502	0.302–20.750	0.395
**Gender (male vs. female)**	2.009	0.758–5.326	0.161			
**Primary tumor location (superior lobes vs. others)**	0.653	0.226–1.889	0.431			
**Brain metastases location (frontotemporal lobes vs. others)**	1.243	0.460–3.361	0.668			
**Tumour size (≤3 cm vs. >3 cm)**	1.552	0.585–4.116	0.377			
**Stage (IV vs. others)**	2.902	0.923–9.128	0.068	4.896	0.890–26.941	0.065
**Adjuvant therapy (yes vs. no)**	1.161	0.538–2.505	0.704			
**Mitotic activity primary tumor (≤15 vs. >15)**	0.716	0.270–1.901	0.502			
**Mitotic activity brain metastases (≤15 vs. >15)**	0.795	0.287–2.202	0.660			
**PI3K primary tumor**	1.008	1.000–1.015	**0.038**			
**PI3K primary tumor; score (≤50 vs. >50)**	7.791	1.718–36.432	**0.008**	0.864	0.058–12.786	0.915
**PI3K primary tumor; score (<60 vs. 60–150 vs. >150)**	2.295	1.135–4.638	**0.021**			
**PINK1 primary tumor**	1.013	1.000–1.026	**0.042**			
**PINK1 primary tumor; score (≤90 vs. >90)**	2.589	0.992–6.759	0.052			
**PINK1 primary tumor; score (<50 vs. 50–100 vs. >100)**	2.236	1.109–4.508	**0.025**	**4.328**	**1.264–14.814**	**0.020**
**PDL1 primary tumor; score (negative vs. positive)**	2.497	0.947–6.585	0.064	3.210	0.768–13.408	0.110

Cox regression model. CI—confidence interval; HR—hazard ratio. Values in bold are statistically significant, *p* < 0.05.

## Data Availability

The data that support the findings of this study are available on request from the corresponding author. The data are not publicly available due to privacy or ethical restrictions.

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
