# Peer review of "PI3K and PINK1 Immunoexpression as Predictors of Survival in Patients Undergoing Resection of Brain Metastases from Lung Adenocarcinoma"

_ijms, 2025, doi:10.3390/ijms26072945_

Round 1
Reviewer 1 Report
Comments and Suggestions for Authors
This manuscript investigates the immunoexpression of PI3K and PINK1 in lung adenocarcinoma patients with brain metastases and their potential prognostic significance. The study uses immunohistochemistry-based H-score assessment and statistical survival analysis to correlate protein expression levels with clinical outcomes.
While this is an interesting study, some serious caveats should be addressed before it is considered for further steps.
Detailed comments are provided below to help the authors improve the manuscript.
- The introduction needs some improvement.
- The introduction repeats well-known lung cancer epidemiology statistics without directly linking them to the study’s hypothesis.
- The manuscript has some grammatical errors and lengthy sections that could be condensed for clarity.
- The study discusses survival analysis but only includes basic Kaplan-Meier plots. Boxplots or scatterplots illustrating PI3K and PINK1 expression across groups would enhance clarity.
- The study is based on only 22 patients, which is insufficient for drawing robust statistical conclusions. There is a high risk of type II errors, and the findings may not be generalizable.
- All patients underwent surgical resection of their brain metastases, which represents a highly selected subset of patients. The exclusion of non-resected cases may introduce significant bias.
- There is no independent validation cohort to confirm the findings. Without validation, the reliability of the identified prognostic biomarkers remains questionable.
- The use of the H-score for immunohistochemistry analysis is acceptable but lacks clear justification for the chosen cutoff values and I don’t understand the logic. Were the thresholds data-driven, or were they arbitrarily set?
- Although it mentions mentions two independent observers, it lacks details about inter-rater agreement (e.g., Cohen’s kappa score) beyond a brief statement on hierarchical kappa tests. This should be quantified and reported.
- The data includes PD-L1 staining but does not provide detailed analysis on its correlation with PI3K or PINK1 expression, so it’s not clear.
- The multivariate model includes only a limited set of variables. Important prognostic factors such as ECOG performance status, extracranial disease burden, and systemic treatment response are missing.
- The study suggests that PI3K and PINK1 could serve as prognostic biomarkers, but there is no direct evidence that these proteins are causally involved in lung adenocarcinoma progression.
- The discussion includes references to other cancers but does not sufficiently compare findings to existing lung adenocarcinoma research. How does PI3K/PINK1 expression compare to known molecular subtypes (e.g., EGFR, KRAS-mutant lung adenocarcinoma)?
- While PI3K inhibitors exist, the manuscript does not explore whether PINK1 is targetable or whether these markers could predict response to therapy.
- Table 1: age range should be provided.
- Limitation section should be clarified and expanded. For example, this sentence needs clarification: “First, there are potential biases in- herent to its retrospective nature.”
- The discussion should explicitly acknowledge the limitations of immunohistochemistry (IHC)-based expression studies and avoid making strong mechanistic claims without supporting experiments (e.g., in vitro or in vivo studies). Example of overstatement: “These findings suggest that PI3K and PINK1 may serve as prognostic biomarkers in lung adenocarcinoma with brain metastases.” This lacks direct proof and should change to: “Our findings indicate a possible association between PI3K/PINK1 expression and survival in lung adenocarcinoma with brain metastases, though further studies are required to confirm their prognostic value and underlying mechanisms.”
- Some questions still remain to be anwered: How does PI3K expression in this study compare with previous reports in lung adenocarcinoma? Is the observed PINK1 expression pattern consistent with or contradictory to other lung cancer studies? Are these findings applicable to EGFR/KRAS/ALK-driven lung adenocarcinoma, which have distinct molecular pathways?
- These points should be elaborated in the Discussion: are there clinically approved PI3K inhibitors that could be relevant for patients with high PI3K expression? Could PINK1 expression be a predictive marker for treatment response, particularly in relation to immunotherapy or chemotherapy resistance? How do the findings impact treatment decisions for lung adenocarcinoma patients with brain metastases?
- One of the most interesting findings in the study is the reduction in PI3K expression in brain metastases compared to primary tumors. However, the discussion fails to explain why this might occur.
- The study reports PD-L1 expression in primary tumors and brain metastases, but the discussion does not connect this to immunotherapy outcomes.
- Since PD-L1 inhibitors (e.g., pembrolizumab) are a standard of care in lung cancer, how do the observed PD-L1 levels compare to response rates?
- Could PI3K/PINK1 expression modulate the tumor immune microenvironment, thereby influencing immunotherapy efficacy?
Improvements are needed.
Reviewer 2 Report
Comments and Suggestions for Authors
I have carefully reviewed this manuscript, which investigates the immunoexpression of PI3K and PINK1 as potential prognostic biomarkers in lung adenocarcinoma with brain metastases. While the study addresses a clinically relevant topic, several critical issues raise concerns about its suitability for publication in its current form.
The most significant limitation is the small sample size (n=22), which is insufficient to draw robust conclusions, especially given the numerous statistical analyses performed. This raises a high risk of false-positive results and overinterpretation of the data. For a study aimed at drawing survival-related conclusions, a much larger patient cohort would be necessary. Additionally, the authors did not include a control group, which could have been useful at this stage of research.
Moreover, the study does not provide a clear comparison with existing literature, making it difficult to assess whether the findings are genuinely novel or merely confirm previously established observations. The reported downregulation of PI3K in metastases compared to primary tumors is an expected finding, and the potential prognostic role of PINK1 has already been explored in other oncological contexts.
The conclusions presented in the manuscript are weak and not adequately supported by the data. While the authors suggest that PI3K and PINK1 could serve as biomarkers, they do not provide a clear rationale on how these markers could be integrated into clinical practice. Additionally, the study lacks external validation, further limiting its impact and applicability.
A further concern is the manuscript’s title, which is grammatically incorrect and poorly formulated. The phrase "with brain resection" is not proper English and reflects a lack of expertise from the authors. A clearer and more precise title should be considered to better reflect the study’s content.
Furthermore, in the discussion, the authors mention predictive algorithms but do not provide sufficient context or references to support this aspect. For example, relevant study that could be cited in this section is PMID: 38759350, but many others could have been mentioned (PMID: 28820304, PMID: 35739135)
For these reasons, I do not consider the manuscript suitable for publication in its current form.
Comments on the Quality of English Languagetitle, as mentioned in the comments
Round 2
Reviewer 1 Report
Comments and Suggestions for Authors
The authors have addressed my comments well.
Author Response
Thank you very much for your feedback. We note that no further action is needed.
Reviewer 2 Report
Comments and Suggestions for Authors
The authors’ response is well-structured and addresses the main points raised in the review. However, some critical issues remain, and the authors' replies are more defensive than truly resolutive in some points. Regarding the small sample size (n=22), their argument that statistically significant results are less likely to be random in small cohorts is questionable without proper correction for multiple comparisons. The lack of a control group remains a limitation, and while they clarify their study population, they do not explain why no comparison group was included or propose an alternative approach to strengthen their findings. Overall, while the response demonstrates a genuine effort to revise the manuscript, some key concerns remain insufficiently addressed, particularly regarding sample size limitations and the absence of a control group.
Author Response
Dear Editor,
Thank you for the helpful comments of the reviewers. All the changes are marked red in the revised manuscript. We hope that the improved version of the manuscript will be accepted for publication.
Response to reviewer 2
1-The authors’ response is well-structured and addresses the main points raised in the review. However, some critical issues remain, and the authors' replies are more defensive than truly resolutive in some points. Regarding the small sample size (n=22), their argument that statistically significant results are less likely to be random in small cohorts is questionable without proper correction for multiple comparisons. The lack of a control group remains a limitation, and while they clarify their study population, they do not explain why no comparison group was included or propose an alternative approach to strengthen their findings. Overall, while the response demonstrates a genuine effort to revise the manuscript, some key concerns remain insufficiently addressed, particularly regarding sample size limitations and the absence of a control group.
We thank the reviewer for their thoughtful and constructive feedback, particularly regarding the limitations related to sample size and the absence of a control group. We have explicitly acknowledged these issues in the manuscript as follows: “Second, the study is limited to a highly specific subset of lung cancer patients, resulting in a small sample size, and we did not evaluate protein immunoexpression in non-metastatic lung adenocarcinomas with comparable characteristics.
We acknowledge that our study involves a relatively small cohort due to the highly specific nature of the patient group, which indeed limits the generalizability of our findings. Although results derived from small cohorts must be interpreted cautiously, we addressed this issue by conducting appropriate statistical adjustments, specifically employing the weighted Bonferroni correction to mitigate the risk of type I errors arising from multiple comparisons.
The novelty of this study lies in the simultaneous analysis of primary tumors and their matched resected brain metastases, rather than comparing metastatic versus non-metastatic tumors. Notably, metastatic tissue was not useful in determining patient prognosis based on PI3K and PINK1 immunoexpression. The intrinsic design focusing on prognostic biomarker evaluation did not initially include a comparison group without brain metastases, which represents an acknowledged limitation. Throughout the manuscript, we have consistently emphasized that the results of this study are limited specifically to this subgroup of patients. We agree with the reviewer that incorporating a comparison group in future research would undoubtedly enhance the interpretative power and allow for more comprehensive conclusions regarding the prognostic significance of PI3K and PINK1 expression. Thank you again for the valuable suggestions, which undoubtedly contribute to improving our manuscript.